# Ingestible hydrogel device

Xinyue Liu[1], Christoph Steiger[2,3], Shaoting Lin[1], German Alberto Parada[1,4], Ji Liu[1], Hon Fai Chan [1,5,6], Hyunwoo Yuk [1], Nhi V. Phan[2], Joy Collins[2], Siddartha Tamang[2], Giovanni Traverso[1,2,3,4] & Xuanhe Zhao [1,7]

Devices that interact with living organisms are typically made of metals, silicon, ceramics, and plastics. Implantation of such devices for long-term monitoring or treatment generally requires invasive procedures. Hydrogels offer new opportunities for human-machine inter-actions due to their superior mechanical compliance and biocompatibility. Additionally, oral administration, coupled with gastric residency, serves as a non-invasive alternative to implantation. Achieving gastric residency with hydrogels requires the hydrogels to swell very rapidly and to withstand gastric mechanical forces over time. However, high swelling ratio, high swelling speed, and long-term robustness do not coexist in existing hydrogels. Here, we introduce a hydrogel device that can be ingested as a standard-sized pill, swell rapidly into a large soft sphere, and maintain robustness under repeated mechanical loads in the stomach for up to one month. Large animal tests support the exceptional performance of the ingestible hydrogel device for long-term gastric retention and physiological monitoring.

[1] Department of Mechanical Engineering, Massachusetts Institute of Technology, Cambridge, MA 02139, USA. [2] The David H. Koch Institute for Integrative Cancer Research, Massachusetts Institute of Technology, Cambridge, MA 02139, USA. [3] Division of Gastroenterology, Department of Medicine, Brigham and Women's Hospital, Harvard Medical School, Boston, MA 02115, USA. [4] Department of Chemical Engineering, Massachusetts Institute of Technology, Cambridge, MA 02139, USA. [5] Institute for Tissue Engineering and Regenerative Medicine, The Chinese University of Hong Kong, Hong Kong, China. [6] School of Biomedical Sciences, The Chinese University of Hong Kong, Hong Kong, China. [7] Department of Civil and Environmental Engineering, Massachusetts Institute of Technology, Cambridge, MA 02139, USA. These authors contributed equally: Xinyue Liu, Christoph Steiger, Shaoting Lin. Correspondence and requests for materials should be addressed to X.Z. (email: zhaox@mit.edu)

The integration of technology with the human body requires devices that are biocompatible, mechanically flexible, and robust over time in biological organisms[1–5]. For instance, devices that reside in the stomach for days to months can enable applications as diverse as in-body physiological monitoring and diagnosis[6,7], bariatric/metabolic interventions[8,9], and prolonged drug delivery[10,11]. Prior approaches to gastric retention include using floating particles on the air/fluid interface[12], and size exclusion through unfolding structures[13,14] or swellable materials[10,15]. Hydrogels represent an ideal material candidate for gastric residency, owing to their inherent similarities to human tissues (e.g., soft, wet, biocompatible, and bioactive)[3,16]. For long-term gastric retention, the ideal hydrogel device needs to swell in the gastric environment from an orally administered pill (diameter of 1.0–1.5 cm) to a size large enough to avoid passing through the pylorus (diameter of 1.3–2.0 cm)[17], and fast enough to avert gastric emptying, which generally occurs 0.5–1.5 h after ingestion[18]. Additionally, the long-acting hydrogel device is required to withstand long-term mechanical forces from the stomach (~1000 cycles per day of 5–10 kPa)[19] and degrade on demand[10]. Tough hydrogels have been used for gastric retention[10], but their low swelling speed presents a challenge for clinical development. Alternative methods to increase the swelling speed include incorporation of interconnected pores into hydrogels[20,21], which tends to adversely affect the hydrogels' swelling ratio and mechanical robustness. Overall, the requirements of high swelling ratio, high swelling speed, and long-term mechanical robustness in the gastric environment are currently not satisfied among existing hydrogels, limiting the applications of hydrogel devices. In nature, *Tetraodontidae* (pufferfish) can rapidly inflate its body into a large and robust sphere when threatened (Fig. 1a)[22,23]. Its fast inflation abilities and mechanical robustness are enabled by the pufferfish's capacity for rapidly imbibing water (no diffusion here) and its stretchable and anti-fatigue skin.

Here, we introduce a pufferfish-inspired hydrogel device, consisting of superabsorbent hydrogel particles that enable the device to quickly imbibe water (instead of diffusion) encapsulated in a soft yet anti-fatigue hydrogel membrane that maintains long-term robustness of the device. The hydrogel device can be ingested as a standard-sized pill (diameter of 1–1.5 cm), rapidly imbibe water and inflate (up to 100 times in volume within 10 min) into a large soft sphere (diameter of up to 6 cm, modulus of 3 kPa), and maintain robustness under repeated mechanical loads over a long time (more than 26,000 cycles of 20 N force over 2 weeks in vitro). We demonstrate the rapid swelling and long-term gastric retention of the hydrogel device for 9–29 days in a large animal model. An embedded sensor in the hydrogel device continuously monitors in-body physiological parameters (here demonstrated by measuring gastric temperature) throughout the retention period. In vitro data suggest that the hydrogel device can also be applied for ultra-long sustained drug delivery. In addition, the hydrogel device is able to shrink on demand to exit the body in response to a biocompatible salt solution. This ingestible and gastric-retentive hydrogel device possesses a set of advantages over conventional ingestible devices made of other materials due to hydrogels' biocompatibility, high water content, and tissue-like softness[1,3,16].

## Results

### Design of the ingestible hydrogel device.
The design of the ingestible hydrogel device is schematically illustrated in Fig. 1. The hydrogel device consists of superabsorbent hydrogel particles (polyacrylic acid, ~450 μm in diameter) encapsulated in an anti-fatigue porous hydrogel membrane (freeze–thawed polyvinyl alcohol, ~750 μm in thickness, ~200 μm in pore diameter) (see Supplementary Figure 1 for biocompatibility data, Supplementary Figure 2 and Methods for fabrication details). This design decouples the swelling ratio, swelling speed, and mechanical robustness of hydrogels. Note that the swelling ratio is defined as $V_{max}/V_0$, and the swelling speed is quantified by the rate constant $k$ in the equation $\partial V/\partial t = k(V_{max} - V)$, where $V_0$ is the initial volume of the hydrogel device, $V_{max}$ is the fully swollen volume, and $V$ is the volume at swelling time $t$[24]. The individual superabsorbent particles can swell ~160 times in volume within 5–10 min (Supplementary Figure 3). As the particles swell, water infiltrates through the pores on the membrane into the hydrogel device, mimicking the rapid imbibition of water by pufferfish. This fast swelling is further facilitated by the capillary effect between particles, which promotes water migration inside the hydrogel device (see Supplementary Note 1 for detailed analysis of swelling). The designed hydrogel membrane of the device is capable of sustaining at least 9000 cycles of 4.3 MPa tensile stress, and thus maintains its robustness under repeated loads, mimicking the anti-fatigue skin of the pufferfish[23]. To enable versatile functionalities such as biosignal recording and extended drug release, wireless sensors and drug depots can be incorporated inside the hydrogel device (Supplementary Figure 4). For safe and on-demand exit of the swollen hydrogel device from the gastrointestinal (GI) tract, a calcium chloride solution that flows into the hydrogel device and deswells the superabsorbent particles can be adopted to induce rapid shrinkage of the swollen hydrogel device (Fig. 1e, f). The calcium concentration is within the safe consumption level[25].

### High-speed and high-ratio swelling of the ingestible hydrogel device.
We first investigated the swelling kinetics of the hydrogel device. As shown in Fig. 2a, a hydrogel device with initial size of 1 cm³ (1 cm in length) swelled into a sphere with a maximum size of 100 cm³ (5.8 cm in diameter) in 10 min in deionized water (pH 7), indicating a high swelling speed and a high swelling ratio (Fig. 2b and Supplementary Movie 1). As shown in the comparison chart in Fig. 2c, the swelling speed of the designed hydrogel device was orders of magnitude higher than that of bulk hydrogels (such as air-dried polyacrylamide and sodium polyacrylate hydrogels, Supplementary Figure 5a) of similar size[10]. The hydrogel device also outperformed the porous hydrogels (such as freeze-dried hydrogels and hydrogel foams, Supplementary Figure 5b) in terms of swelling ratio[20]. By varying Young's modulus of the polyvinyl alcohol hydrogel membrane, we tuned the swelling ratio of the hydrogel device while maintaining its high swelling speed (Fig. 2d) (see Supplementary Note 1 and Supplementary Figure 6 for detailed analysis). We further performed swelling tests of the hydrogel device with a membrane modulus of 3 kPa in porcine gastric fluid and simulated gastric fluid (SGF, pH 3). The hydrogel device swelled ~25 times in both media within 10 min, indicating that a hydrogel device with initial size of 3 cm³ (1.4 cm in length) is capable of swelling to 75 cm³ (5.2 cm in diameter) within 10 min (Fig. 2e, Supplementary Figure 7, and Supplementary Movie 2). Figure 2f, g summarizes the swelling ratios and swelling speeds of the hydrogel device with various membrane moduli in water, SGF (pH 3), and porcine gastric fluid. Given that the hydrogel device's swelling is faster than the typical gastric emptying time[18], its initial dimension is less than the diameter of the esophagus[26], and its swollen size is greater than the diameter of the pylorus[17], we conclude that the hydrogel device is compatible with oral administration and potential gastric residence.

To introduce a rescue strategy for potential complications caused by ingestible devices in the GI tract (e.g., bowel

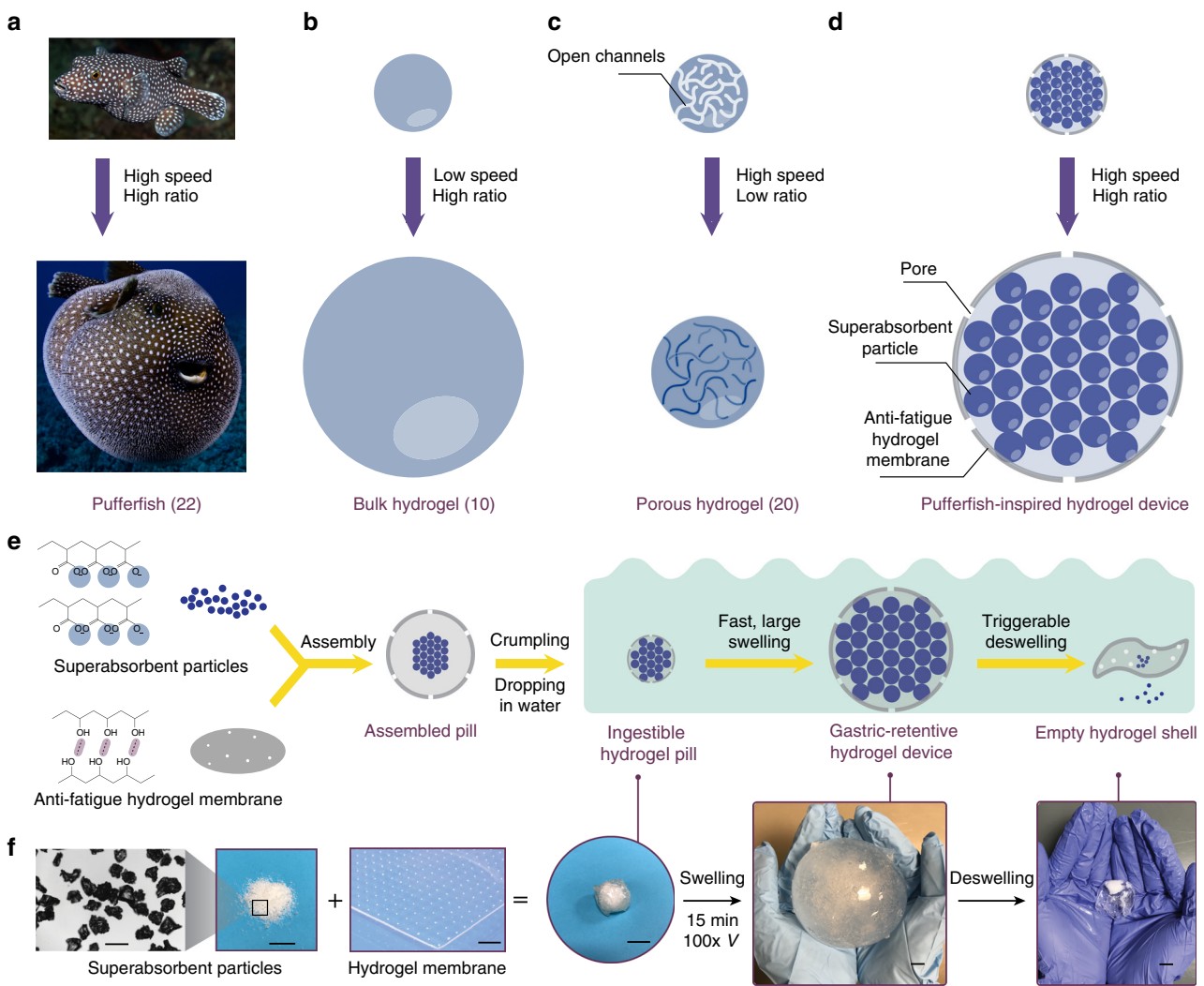

**Fig. 1** Design of the pufferfish-inspired ingestible hydrogel device. **a** A pufferfish inflates its body into a large ball by rapidly imbibing water[22]. **b** Bulk hydrogels swell in water with a low swelling speed[10]. **c** Porous hydrogels swell in water with a low swelling ratio[20]. **d** The designed hydrogel device swells in water with both a high speed and a high ratio. **e** Schematic of the fabrication process and working principle of the designed hydrogel device. **f** Photographs of the fabrication process and working principle of the designed hydrogel device. Scale bars are 0.5 mm for the first image in (**f**) and 10 mm for the other images in (**f**)

obstruction)[27,28], we demonstrated the rapid shrinkage of the swollen hydrogel device by introducing calcium ions, which flowed into the hydrogel device and binded with the carboxyl groups in the polyacrylic acid hydrogel. As shown in Supplementary Figure 8a–c and Supplementary Movies 3 and 4, the calcium solution (0.6 M) induced the deswelling of super-absorbent particles within 15 min, leading to shrunken particles in an empty hydrogel shell. The hydrogel device is designed such that its shrunken state is small and compliant (~1 cm in diameter when compacted, 3–47 kPa in Young's modulus) in order to safely pass through the GI tract[13,17]. The amount of calcium needed to trigger the shrinkage of hydrogel device (0.6 M; 2.1 g for a typical stomach with 87 mL gastric fluid) is less than the tolerable upper intake level of calcium (2.5–3 g per day)[25], but approximately 20 times greater than the amount in calcium-rich foods (for example, 0.03 M in milk)[29]. Additionally, we demonstrated that a low concentration of calcium (0.03 M) did not significantly affect the SGF-swollen hydrogel device (Supplementary Figure 8d), supporting the stability of the hydrogel device in the stomach against the regular calcium intake.

**Mechanical softness and robustness of the ingestible hydrogel device.** To provide a mechanically flexible and conformable interface with the stomach, the hydrogel device was designed such that the overall moduli ranged from 3 to 10 kPa with membrane moduli of 3–47 kPa (Fig. 3 and Supplementary Figure 9). The hydrogel device is much softer than most existing ingestible devices which contain dry, rigid, and non-degradable materials in order to maintain their structural integrity within the GI tract and to protect electronic components from the harsh environment, such as stomach acid (Fig. 4b)[6,7,13,14,27]. The compliance of the hydrogel device is in line with that of common foods (e.g., 15 kPa for noodles[30] and 24 kPa for tofu[31,32]) and human tissues (e.g., 7 kPa for muscle and 85 kPa for skin[33]). The food and tissue-level softness of the hydrogel device alleviates the potential of GI mucosa injury.

To ensure the long-term robustness of the hydrogel device in the gastric environment, we evaluated its mechanical performance, including the membrane material and the overall device. The stomach generates hydrodynamic flows and cyclic compressive forces in order to grind food into smaller particles, mix them with gastric fluid, and empty them through the pylorus[19].

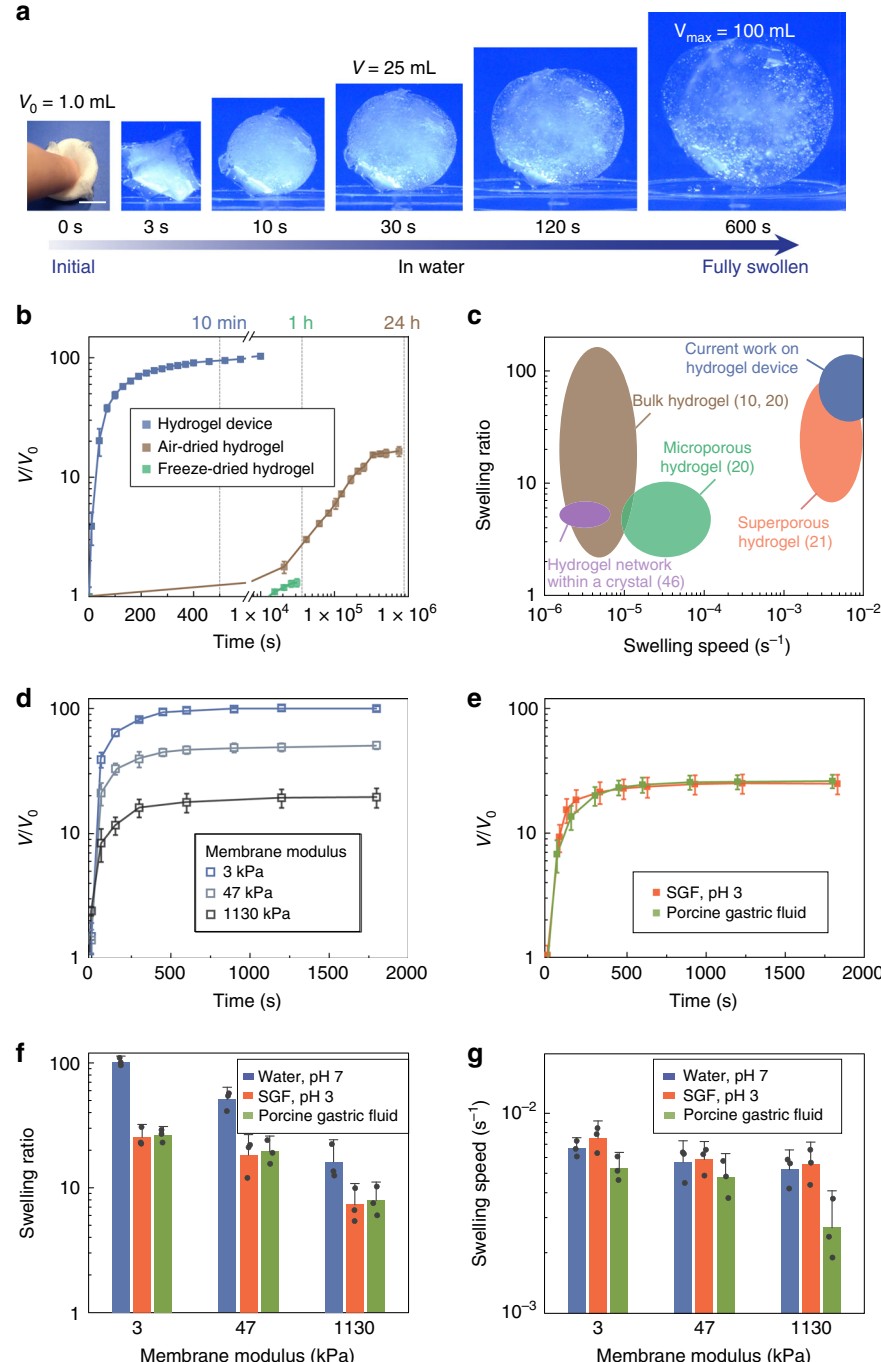

**Fig. 2** High-speed and high-ratio swelling of the ingestible hydrogel device. **a** Time-lapse images of the hydrogel device swelling in water (pH 7). **b** Volume changes of the hydrogel device (membrane modulus 3 kPa), air-dried hydrogel, and freeze-dried hydrogel of the same size as a function of swelling time in water. **c** Comparison of the swelling ratios and speeds in water between the hydrogel device in current work and previously reported hydrogels[10,20,21,46]. **d** Volume changes of the hydrogel devices with various membrane moduli as functions of swelling time in water. **e** Volume changes of the hydrogel devices (membrane modulus 3 kPa) as functions of swelling time in porcine gastric fluid and SGF (pH 3). **f** Swelling ratios of the hydrogel devices with various membrane moduli in water, SGF (pH 3), and porcine gastric fluid. **g** Swelling speeds of the hydrogel devices with various membrane moduli in water, SGF (pH 3), and porcine gastric fluid. Scale bars are 10 mm in (**a**). Data represent the mean ± s.d. ($N = 3$)

However, most hydrogels are susceptible to fatigue failure under cyclic mechanical loads, especially in acidic environments[34]. We found that freeze–thawed polyvinyl alcohol hydrogels can be used as the anti-fatigue membrane for the hydrogel device under cyclic mechanical loads (see Methods for details on preparation of the membrane). The freeze–thawing treatment introduced nano-crystalline domains into the polyvinyl alcohol hydrogel, making it strong, tough, and fatigue resistant while maintaining a low

modulus[34,35] (Fig. 3 and Supplementary Figures 10 and 11). After being immersed in SGF (pH 3) at body temperature (37 °C) for over 2 weeks, the hydrogel membrane demonstrated high strength of over 7 MPa (Fig. 3b) and high toughness of over 1000 J m$^{-2}$ (Fig. 3c). The hydrogel membrane was also capable of sustaining 9000 cycles of 4.3 MPa tensile stress, and thus maintained the robustness of the swollen hydrogel device under repeated loads (Supplementary Figure 10).

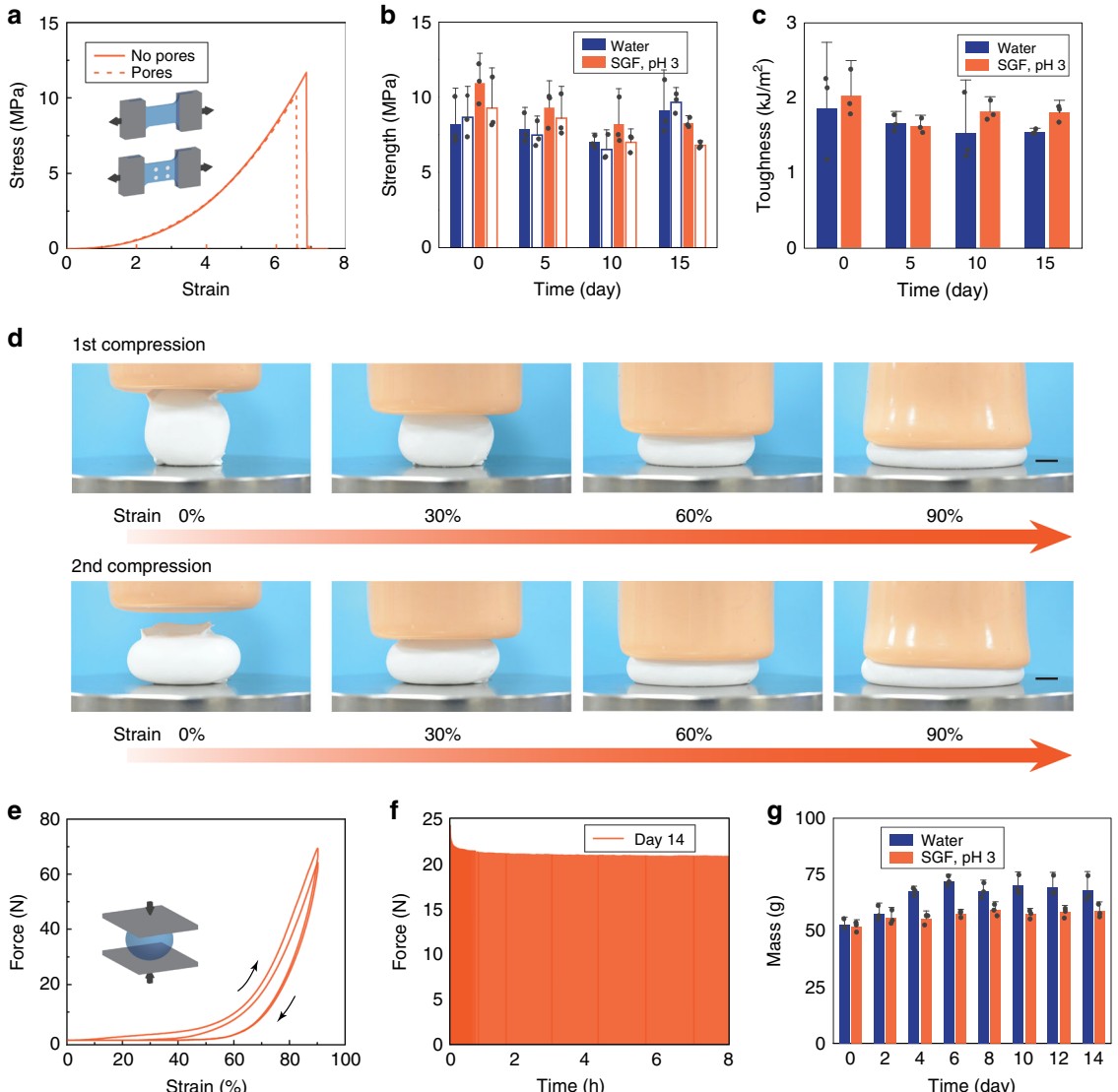

**Fig. 3** Mechanical robustness of the ingestible hydrogel device. **a** True stress–stretch curves of the polyvinyl alcohol hydrogel membranes with and without pores, which have been immersed in SGF (pH 3) at 37 °C for 12 h. **b** Tensile strength of the hydrogel membranes with (open) and without (filled) pores, which have been immersed in water or SGF (pH 3) at 37 °C for 0–15 days. **c** Fracture toughness of the hydrogel membranes, which have been immersed in water or SGF (pH 3) at 37 °C for 0–15 days. **d** Time-lapse images of an SGF (pH 3)-saturated hydrogel device (diameter ~3.6 cm at undeformed state) exposed to a maximum compressive force of 70 N and a strain of 90%. **e** Force–strain curves of the SGF (pH 3)-saturated hydrogel device exposed to a maximum compressive force of 70 N and a strain of 90% for two cycles. **f** Measured compressive forces applied to a hydrogel device (diameter ~4.8 cm at undeformed state) on day 14 (the hydrogel device was immersed in SGF (pH 3), and sustained 1920 cycles of 40% compressive strains for 8 h per day). **g** Measured mass of the hydrogel device after 1920 cycles of 40% compressive strain for 8 h per day over 14 days. Scale bars are 10 mm in (**d**). Data in (**b**, **c**, **g**) represent the mean ± s.d. (N = 3)

Furthermore, we validated the high robustness of the swollen hydrogel device under mechanical loads. We showed that the hydrogel device (diameter ~3.6 cm, swollen in SGF) could sustain large compressive strains up to 90% and high forces up to 70 N (Fig. 3d, e, Supplementary Figure 9, and Supplementary Movies 5 and 6). Considering the dimension of the hydrogel device, the effective compressive stress was calculated as ~70 kPa, which is much higher than the maximum gastric pressure (i.e., ~10 kPa)[19]. In addition, we applied 1920 cycles of 40% compressive strain on a hydrogel device (diameter ~4.8 cm) in SGF (pH 3) for 8 h every day (i.e., 26,880 cycles in total for 14 days). The steady-state compressive force reached 20 N over 14 days, corresponding to an effective compressive stress of ~10 kPa (Fig. 3f). We also recorded the mass of the swollen device after cyclic compression every day, and no mass loss was detected over 2 weeks (Fig. 3g).

In contrast, an alternative hydrogel device made of a tough hydrogel membrane but with short fatigue life (polyacrylamide-agar hydrogel, Supplementary Figure 10c) showed severe softening and loss of mass after 1920 cycles of 40% compressive strain on the first day (Supplementary Figure 11).

**Long-term gastric retention and physiological monitoring of the ingestible hydrogel device.** Having demonstrated the significant swelling performance, mechanical softness, and long-term robustness in vitro, we subsequently tested the gastric retention of the hydrogel device in a Yorkshire pig model (30–50 kg in weight, the number of replicates N = 3 per group). Figure 4a illustrates the working principle of the hydrogel device in the GI tract. The ingestible hydrogel device enters through

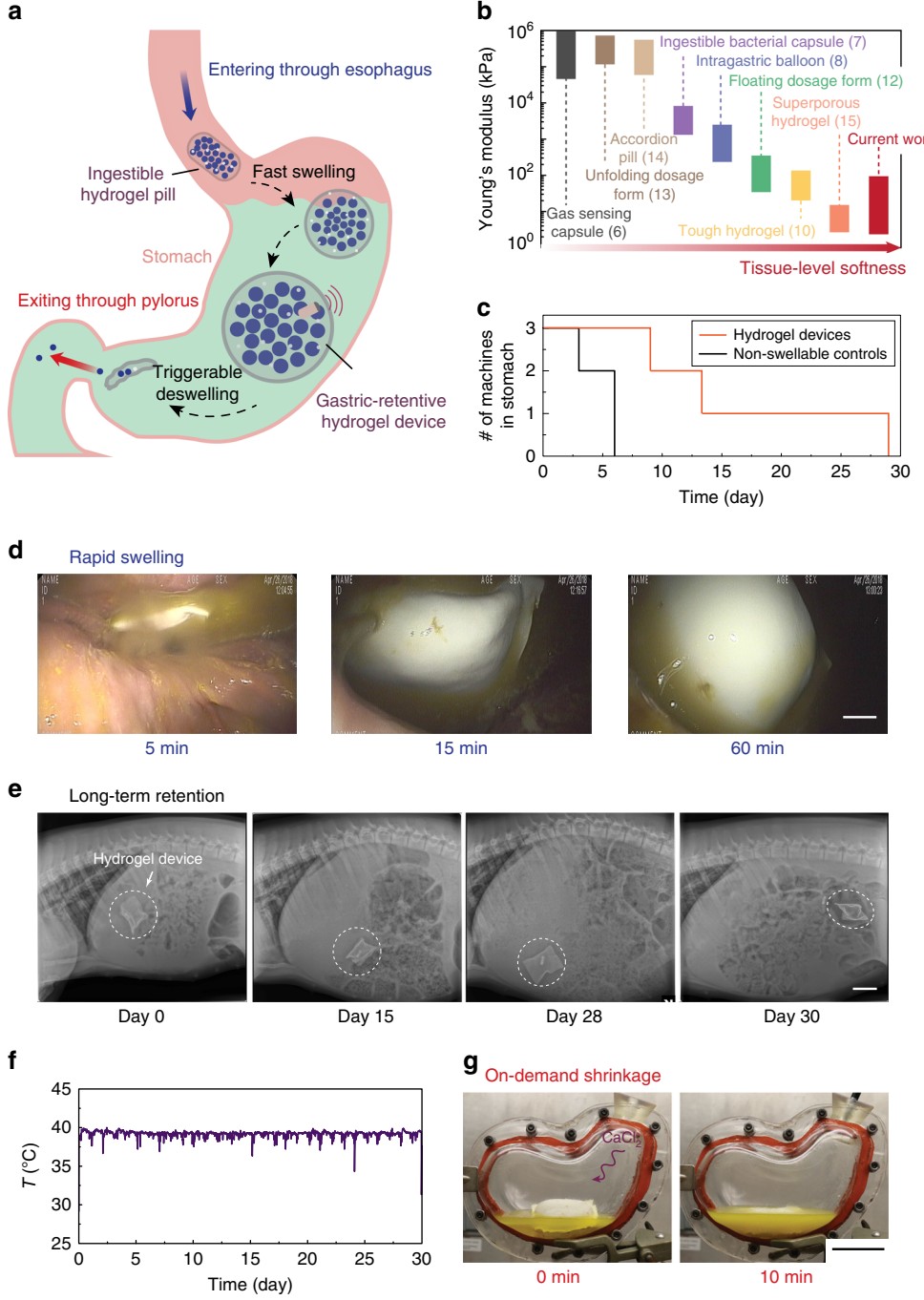

**Fig. 4** Long-term gastric retention and physiological monitoring of the ingestible hydrogel device. **a** Working principle of the gastric-retentive hydrogel device, which enters through the esophagus into the stomach as an ingestible pill, resides in the stomach in its swollen state for a prolonged period of time, and exits through the pylorus as a shrunken capsule and small particles. **b** Comparison of Young's moduli among recently reported ingestible devices[6–8,10,12–15] and the hydrogel device in current work. **c** Number of hydrogel devices and non-swellable devices (i.e., without any superabsorbent particles) being retained in the porcine stomach as a function of time ($N = 3$ for each group). **d** Endoscopic images depicting the swelling of the hydrogel device in the porcine stomach. **e** X-ray images of the hydrogel device residing in the porcine stomach before being emptied into distal parts of the GI tract (here shown for 29 days in stomach). **f** Continuous measurement of porcine gastric temperature by a sensor embedded in the hydrogel device. **g** Photos of ex vivo shrinkage of the hydrogel device triggered by the addition of 0.6 M calcium chloride solution. Scale bars are 10 mm in (**d**), 5 cm in (**e**) and (**g**)

the esophagus into the stomach where it resides in its swollen state for a prolonged period of time. As depicted in Fig. 4d and Supplementary Figure 12, the hydrogel device with the initial size of ~3 cm³ (diameter of 1 cm and length of 3 cm, compatible with oral administration) absorbed gastric fluid and swelled to ~50 cm³ within 60 min in the porcine stomach. Radiographic data suggested that it retained its swollen shape and dimensions

in the peristaltic and contractile stomach without being evacuated through the pylorus for a long time (9–29 days; Fig. 4c, e and Supplementary Table 1). During its residency in the stomach, the hydrogel device floated freely with no radiographic or clinical evidence of bowel obstruction. In contrast, control samples of the non-swellable hydrogel device with no superabsorbent particles (but otherwise identical design to the original hydrogel device)

had a much shorter gastric residence time (3–6 days) in the porcine stomach, indicating that high swelling ratio of the hydrogel device is required for long-term gastric retention (Fig. 4c and Supplementary Table 1). One potential limitation of our in vivo tests is that pigs have slower gastric emptying than humans[18,36]. Also, the gastric compression force in pigs is slightly lower in pigs than that in humans[19,37]. For the successful translation to humans, further testing in other large animal species such as dogs will likely be required[38].

To reveal the hydrogel device's potential applications as a prolonged platform in GI tract to carry functional elements, a temperature sensor (DST nanoRF-T, Star-Oddi) was embedded in the hydrogel device. The temperature of the porcine stomach was recorded for 29 days (and the entire GI tract for 30 days, Fig. 4f), revealing the capability to monitor in-situ physiological signals for an extended period of time. We replotted the temperature profiles in Fig. 4f on a daily basis in Fig. 5a,

exhibiting distinct features of day–night cycles. The gastric temperature is known to be linked with the food/drink intake pattern, that is, the consumption on cooler food/drink results in a gastric temperature variation characterized by a sudden drop and subsequent rise[39]. As shown in a detailed temperature profile on day 17 (Fig. 5b), there were different phases with different ingestion activities happening in 1 day. The pig gastric temperature stabilized at 39.2 °C when the pig was asleep (from 12:00 am to 6:30 am). When the pig was awake (from 6:30 am to 12:00 am), the temperature mostly showed small or moderate fluctuations (39.6–38.5 °C), possibly indicating small or medium amounts of continuous food/drink ingestion. In addition, a sharp temperature drop from 39.2 °C to 37.2 °C within 20 min possibly revealed the massive food/drink intake starting from 11:30 am.

The recorded gastric temperature over a long period of time can be used to characterize the dietary habit of the subject[39,40]. In order to visually represent the temperature pattern, the heatmaps

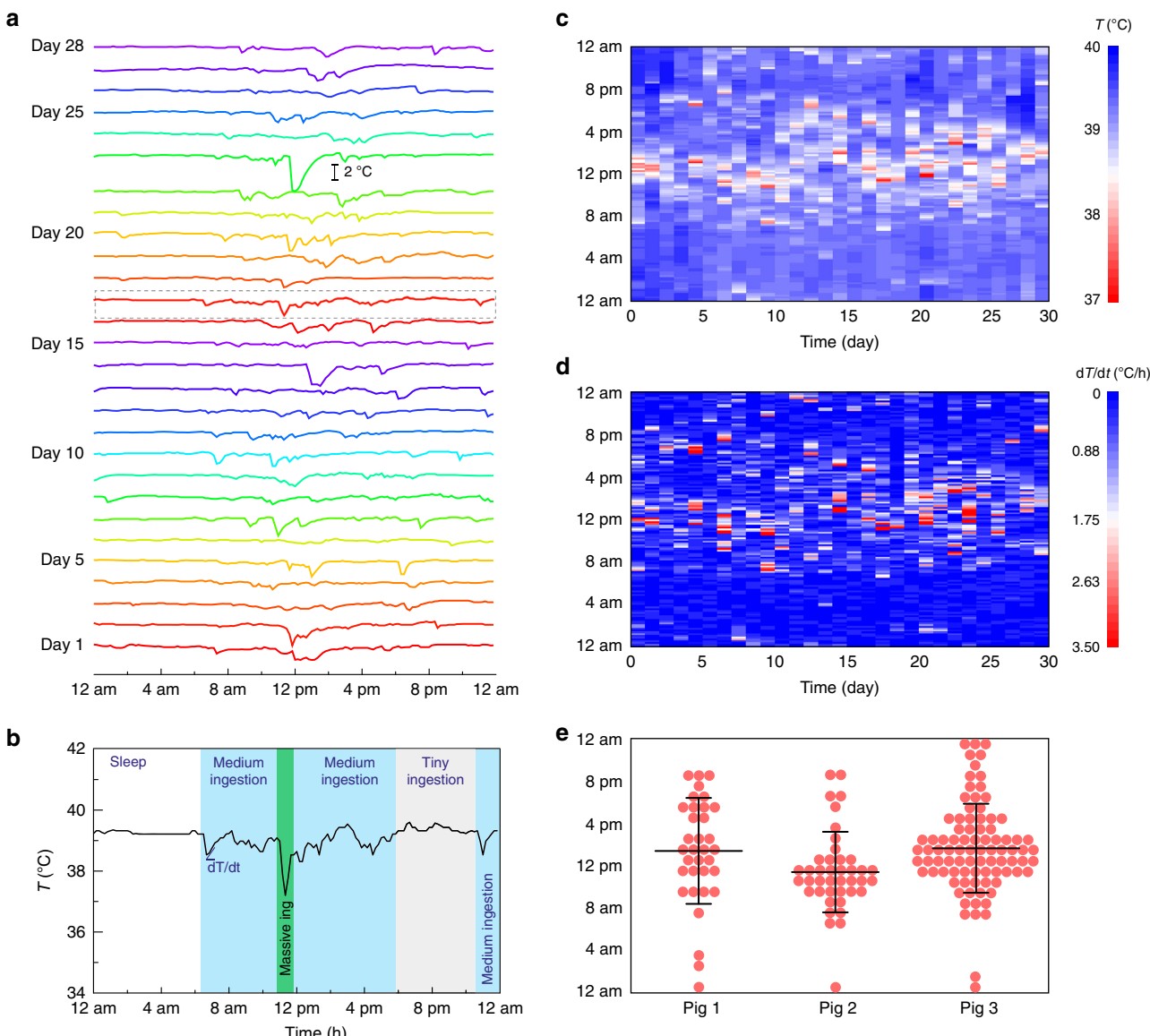

**Fig. 5** Analysis on the prolonged porcine gastric temperature profile measured. **a** The long-term measured gastric temperature in Fig. 4f are replotted on a daily basis. **b** On a single day (day 17), the temperature profile is divided into different phases with different ingestion activities based on the degree of fluctuations. **c** The heatmap of the temperature (T) measured by hydrogel device over 29 days. **d** The heatmap of the absolute temperature derivative ($|dT/dt|$) measured by hydrogel device over 29 days. **e** The time slots (1 h) with food intake (if any $|dT/dt| > 1.75$ during the time slot) are marked as events for different pigs. Data represent the mean ± s.d. N = 34 events for pig 1 in 9 days, N = 41 events for pig 2 in 13 days, and N = 88 events for pig 3 in 29 days

of the temperature ($T$) and the absolute of temperature derivative ($|dT/dt|$) are graphically plotted in Fig. 5c, d as functions of the day and hour. It can be seen that most of the temperature fluctuations occurred between 8 am and 8 pm on different days, indicating the regular dietary habit of the pigs was kept within 1 month. We then assumed that there existed food ingestions when $|dT/dt| > 1.75$ (the middle point defined in the heatmap color scale) during 1 h, otherwise no food intake. In Fig. 5e, a time slot (1 h) with food intake was marked as an event in terms of three pigs we used. The average meal times and standard deviations (s.d.) for three pigs were calculated to be 1:25 pm ± 5.0 h (mean ± s.d., $n = 34$ for 9-day recording, where $n$ is the number of events), 11:25 am ± 3.8 h ($n = 41$ for 13-day recording), and 1:40 pm ± 4.2 h ($n = 88$ for 29-day recording). The temperature patterns of three pigs had some common characteristics; for example, 8 am–8 pm was the most active food intake period for all three pigs. The pigs also showed diversity in the ingestion time. For instance, pig 1 had more uniformly distributed meal time (s.d. ~5 h), and pigs 2 and 3 had more concentrated food intake pattern around noontime (s.d. ~4 h). The information we obtained from the prolonged biosignal measurement may contribute to the understanding of the GI environment, and potentially monitoring of the behavior pattern, analysis of the circadian rhythm, and diagnosis of abnormality[40].

Additionally, we demonstrated the ex vivo triggerable shrinkage of the hydrogel device for easy and clear visualization. As shown in Fig. 4g and Supplementary Figure 12, calcium ions (0.6 M) induced deswelling of the hydrogel device within 10 min in a transparent plastic stomach model containing porcine gastric fluid. On-demand shrinkage of the hydrogel device not only provides a rescue strategy for potential device-related GI tract obstructions, but also serves as an approach to tailorable residence time (e.g., when physiological monitoring is no longer needed).

## Discussion

Recently developed soft ingestible hydrogels (Fig. 4b) are limited by their mechanical weakness, low swelling ratio, and/or low swelling speed, leading to short gastric residence time (Supplementary Table 2)[10,12,15]. The current work represents a noninvasive, self-contained GI-resident device almost entirely made of hydrogels that can be orally administrated as a small pill, swell in the stomach within 60 min, and remain soft but robust in the gastric cavity for up to 29 days. To our best knowledge, this is the first design of a hydrogel that achieves superior properties of high swelling ratio, high swelling speed, and long-term robustness simultaneously. In addition, we report an ingestible and gastric-retentive hydrogel device capable of 1-month continuous measurement of gastric temperature in a pig model, which has never been achieved before.

Besides the applications of in vitro drug release and in vivo temperature sensing we have shown, our hydrogel device can be potentially used as a versatile platform for other functions that closely interact with the digestive system in the human body. For example, long-term measurements of other biosignals may be enabled by the miniature sensors (for motion, salinity, pressure, pH, gas, or other biomarkers) embedded in the hydrogel device. Moreover, the long-acting hydrogel device may be used for monitoring medication-taking patterns, visualizing GI tract disorders by carrying a mini camera, modulating gut microbiota via the addition of probiotics/prebiotics, and inducing satiety to control obesity by volume exclusion. Beyond that, by decoupling the mechanical and swelling properties, our hydrogel design supports the possibility of achieving osmotic-driven high-force and high-speed actuation, potentially opening avenues to

applications for hydrogel-based biomedical devices[41] and soft robotics[42].

## Methods

**Synthesis of the hydrogel membranes**. All types of anti-fatigue hydrogel membranes were prepared from polyvinyl alcohol powders (PVA; Mw 146,000–186,000, 99+% hydrolyzed, Sigma-Aldrich). An aqueous solution of 10 wt % PVA was dissolved by stirring at 75 °C for 6 h, and mixed and defoamed by using a centrifugal mixer (AR-100, Thinky) for 1 min. The solution was cast in a 0.6-mm-thick custom-made glass mold, frozen at −20 °C for 8 h, and thawed at 25 °C for 3 h. The 8-h-freezing and 3-h-thawing was defined as one freeze–thaw cycle. Samples that underwent one freeze–thaw cycle were the soft hydrogel membranes (Young's modulus 2.6 kPa), and samples that underwent four freeze–thaw cycles were the medium hydrogel membrane (Young's modulus 47 kPa). The PVA hydrogels, after four freeze–thaw cycles, were further air-dried at 37 °C for 1 h, and annealed at 100 °C for 1 h, so that we obtained the stiff hydrogel membranes (Young's modulus 1.13 MPa). For radiographic visualization in vivo, 23 wt% of radio-opaque barium sulfate (Sigma-Aldrich) was incorporated in the hydrogel membrane. The addition of barium sulfate did not affect the mechanical properties as suggested by identical stress–strain curves and moduli (~46 kPa for both). All types of resultant PVA hydrogels were left in the molds for 2 days at room temperature before further use.

An alternative hydrogel membrane was prepared using a polyacrylamide-agar tough hydrogel. Agar (2 g; Sigma-Aldrich), acrylamide (18 g; Sigma-Aldrich), Irgacure 2959 (0.284 g; Sigma-Aldrich) as the photoinitiator, and N,N′-methylenebisacrylamide (0.007 g; Sigma-Aldrich) as the crosslinker were dissolved in 75 mL water at 90 °C. The solution was degassed thoroughly and poured in the custom-made glass mold (0.6 mm thick). The precursor solution was allowed to cool in the mold at room temperature to form the agar network, and then exposed to UV irradiation (365 nm wavelength, 8 W) for 1 h to form the polyacrylamide network. Unreacted monomer and photoinitiators are leached out from the membrane for 2 days by deionized water.

**Fabrication of the hydrogel devices**. For water permeation into the hydrogel device, the hydrogel membranes were introduced with uniform pores (~200 μm in diameter, two pores per cm²) using laser cutting (Epilog, Supplementary Figure 2). The hydrogel membranes were then trimmed based on a Parafilm template (Bemis) and assembled into a pocket or cube structure for subsequent loading with a specific amount of superabsorbent particles (sodium polyacrylate homopolymers; Waste Lock 770, M2 Polymer Tech). The edges of the assembled pocket or cube structure were adhered using biocompatible ethyl cyanoacrylate glue (Loctite). After the Parafilm template was removed, the hydrogel device was further crumpled into a pill size.

**Swelling tests**. The prepared hydrogel devices were submerged in aqueous media for swelling tests, which included water, SGF (pH 3), or porcine gastric fluid. Compendial SGF was prepared with sodium chloride (150 mM) and hydrochloric acid (1 mM) in water[13,43]. Porcine gastric fluid was withdrawn endoscopically when we performed another observational endoscopic study in the pig, and stored at −80 °C. During swelling studies, the increase of volume over time was monitored using a DSLR camera (Nikon D7000), and the mass of hydrogel devices was monitored using analytical balance (Denver Instrument). Swelling of superabsorbent particles was recorded using microscopy (Nikon Eclipse LV100ND). The volume at each time point was obtained by the area to the 1.5th power, where the area was accessible from the time-lapse images. The volume change was expressed as $V/V_0$, normalized by the initial volume $V_0$.

**Swelling of bulk and porous hydrogels**. The air-dried bulk hydrogel and the freeze-dried porous hydrogel were prepared from polyacrylamide-agar following the same procedure described for synthesis of the hydrogel membranes (see above) with modifications. Those modifications included that the as-prepared hydrogels were dried at room temperature for 2 days to form bulk hydrogels. To prepare porous hydrogels, the hydrogels were fully swollen in water for 2 days, frozen at −20 °C for 1 day, and lyophilized for 3 days. The mass of each hydrogel was recorded over time during their swelling. The measured swelling times of the air-dried and freeze-dried hydrogels were normalized to the initial sample size of 5 mm.

**Deswelling test**. Deswelling characteristics of the hydrogel devices were evaluated by inducing deswelling with 0.6 or 0.03 M calcium chloride in swelling media of water or SGF (pH 3). The mass and volume of the hydrogel devices were recorded over time in the deswelling tests following the procedures described in the swelling tests (Supplementary Figure 8).

**Mechanical testing of the hydrogel membranes**. To assess the mechanical properties of hydrogel membranes under simulated physiological conditions, hydrogel membranes with or without pores were incubated in various media of water (pH 7) and SGF (pH 3) at 37 °C before performing mechanical testing. They were cut into dog-bone specimen with 6.5 mm in width, 15 mm in gauge length,

and 0.75 mm in thickness for all samples, and 1.2 mm in crack length for notched samples. At 12 h, 5 days, 10 days, and 15 days after incubation, true stress–strain curves of the unnotched hydrogel membranes were measured using a mechanical testing device (Z2.5, Zwick-Roell) with a 20 N load cell. The strain rate was imposed to $2 \, s^{-1}$.

The true stress $\sigma$ was calculated by $\sigma = F(1 + \varepsilon)/(Wt)$, where $F$ is the measured force, $W$ is the sample width, $t$ is the sample thickness, and $\varepsilon$ is the measured nominal strain[44]. The maximum stress that was reached was identified as tensile strength of the hydrogel. To measure the fracture toughness, we first performed uniaxial tension tests on an unnotched sample, and calculated the energy density by $W(\varepsilon) = \int_{1}^{1+\varepsilon} S d\varepsilon$, where $S = F/(Wt)$ is the measured nominal stress. Thereafter, we performed single-notched tensile tests on the notched sample. The fracture toughness was calculated to be $\Gamma = 2k(\varepsilon) \cdot c \cdot W(\varepsilon)$, where $k$ is a slowly varying function of the applied stretch expressed as $k = 3/\sqrt{1 + \varepsilon}$, $c$ is the crack length at undeformed configuration, and $W$ is the strain energy density measured in the unnotched sample[44].

To test the fatigue properties of hydrogel membranes, we performed cyclic tensile loading on the hydrogel membranes with pores in a water bath (pH 7) at 25 °C with a benchtop mechanical tester (UStretch, CellScale), using a 44 N load cell. Forces applied were recorded over time. The strain rate was imposed to $5 \, s^{-1}$.

**Mechanical testing of the hydrogel devices**. The hydrogel devices, which were fully swollen in SGF (pH 3) or water for 1 h beforehand, were exposed to cyclic compression using a mechanical testing device (Z2.5, Zwick-Roell) with a 2500 N load cell and a cylindrical soft indenter (Ecoflex, diameter 70 mm). The strain rate imposed was $2 \, s^{-1}$. The maximum engineering strain was 40% for long-time compression, and the hydrogel devices (diameter ~4.8 cm, maximum cross-section area ~20 $cm^2$ at undeformed state) in the medium underwent 8-h cyclic loading every day. In the short-run test, the maximum engineering strain was 90%, and the devices (diameter ~3.6 cm, maximum cross-section area ~10 $cm^2$ at undeformed state) in the air underwent two cycles. The effective compressive stress was calculated by dividing the compressive force by the maximum cross-section area of the undeformed hydrogel device. We used the Hertz model to obtain the effective moduli of whole hydrogel devices from the compression curves[45].

**Cytotoxicity analysis**. Cell viability was tested on Caco-2 cells (American Type Culture Collection). Caco-2 cells (clone: C2BBe1, passage 48–58) were cultured in Dulbecco's modified Eagle medium (Life Technologies) supplemented with 10% fetal bovine serum (Sigma-Aldrich), 1× non-essential amino acids solution (Life Technologies), 1× GlutaMAX (Life Technologies), and penicillin/streptomycin (Life Technologies). 5 mL of fresh culture media were introduced to the hydrogel device and its components, including the hydrogel membrane and superabsorbent particles, for 1 day at 37 °C (hereafter referred as pre-exposed medium). One day after the Caco-2 cells were seeded, the culture medium was replaced with the pre-exposed medium, and the cells were co-incubated with the pre-exposed medium for 72 h without changing the medium. Cells treated with 70% ethanol and untreated cells were used as a negative control and positive control, respectively. Finally, cell viability was analyzed using a commercial assay according to the manufacturer's protocol (LIVE/DEAD viability/cytotoxicity kit for mammalian cells, Life Technologies). The cells were imaged using a Leica SP8 upright confocal microscope.

**In vitro temperature sensing of the hydrogel devices**. The temperature sensor (DST nanoRF-T, Star-Oddi), 1.5 cm in length and 0.5 mm in diameter was embedded in the ingestible hydrogel device, which was then allowed to swell in water or SGF (pH 3). The entire hydrogel device in media was placed in an incubator set to 37 °C, or alternatively placed at 37 °C and at room temperature. The antenna was mounted on the wall of the incubator, enabling real-time temperature reading in the hydrogel device.

**In vitro drug release of the hydrogel devices**. Formulations containing 2.5 mg caffeine (Sigma-Aldrich), 0.14 g pluronic P407 (Sigma-Aldrich), and 1 g linear polycaprolactone (PCL; Mw 45,000, Sigma-Aldrich) were combined, melted at 90 °C, and mixed vigorously. The molten mixture was transferred into a small acrylic mold 1.5 cm in length and 0.5 mm in diameter, heated at 90 °C for 2 h, and air-cooled to room temperature. The PCL drug depot was then incorporated into the ingestible hydrogel devices. Caffeine release from the hydrogel device was monitored using UV–vis spectrometry at 275 nm (BioMate 3S, Thermo-Fisher).

**Ex vivo study in a stomach model**. A stomach model custom-made of plastics and containing 75 mL porcine gastric fluid was used as an accessible ex vivo model. To trigger the deswelling of the fully swollen hydrogel device, 7.5 mL calcium chloride solution (50 wt%) was added 30 min after the insertion and swelling of the hydrogel device in gastric fluid. A snare catheter (Captivator II Single-Use Snare, Boston Scientific) was used to retrieve the shrunken hydrogel device through the opening. The process was monitored using a camera (EOS 70D, Canon, Supplementary Figure 12).

**Porcine in vivo model**. All procedures were conducted in accordance with protocols approved by the Massachusetts Institute of Technology Committee on Animal Care. Six separate female Yorkshire pigs weighing approximately 30–50 kg were used for in vivo evaluation (experimental group: 39.7 ± 2 kg; control group: 40.3 ± 8 kg). Following overnight fasting, the animals were sedated with Telazol (tiletamine/zolazepam) 5 mg $kg^{-1}$, xylazine 2 mg $kg^{-1}$, and atropine 0.04 mg $kg^{-1}$, followed by endotracheal intubation and maintenance anesthesia of isoflurane (1–3% in oxygen). The barium sulfate-labeled ingestible hydrogel device comprising a temperature sensor (DST nanoRF-T, Star-Oddi) was placed in the stomach using an esophageal overtube (US Endoscopy) with endoscopic guidance. The swelling of the hydrogel device in the stomach was visualized endoscopically at 5, 15, and 60 min after administration. Temperature data were measured every 10 min. The non-swellable hydrogel device with a temperature sensor but containing no superabsorbent particles inside was delivered as the control experiment. Radiographs were performed every 48–72 h to monitor the integrity and transit of the devices as well as any radiographic evidence of bowel obstruction. Furthermore, all animals were monitored clinically at least twice a day for any evidence of morbidity, including lethargy, inappetence, decreased fecal output, abdominal distension, and vomiting. The temperature sensor in the hydrogel device was retrieved from the pigs' feces after exiting from the GI tract. The temperature data were recorded every 10 min in porcine stomach with resolution of ±0.032 °C.

## Data availability
The preclinical data and gastric temperature profiles can be accessed from the dropbox [https://www.dropbox.com/sh/gavtugf3khinaaz/AAAKEksfw7r-t11fm1Ooa_dha?dl=0]. Requests for other materials should be addressed to the corresponding author.

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

## Acknowledgements

The authors thank R. Langer for help with the preclinical trial design, S. Zhang, D. Larson, E. Strobach, and J. Keough for helpful discussion, and A. Hayward, V. Soares, T. Hua, and H. Sun for help with the in vivo porcine work. This work is supported by National Science Foundation (CMMI-1661627), National Institutes of Health (EB000244), and Bill and Melinda Gates Foundation (OPP1139937 and OPP1139921). C.S. acknowledges support from the Alexander von Humboldt Foundation (Feodor Lynen Fellowship). H.Y. acknowledges financial support from a Samsung Scholarship. G.T. acknowledges support from the Division of Gastroenterology, Brigham and Women's Hospital.

## Author contributions

X.L., S.L., and X.Z. conceived the idea of the ingestible hydrogel device for gastric retention and designed the study in vitro. C.S. and G.T. designed the preclinical in vivo study and adapted the hydrogel device for psychological monitoring. X.L., S.L., G.A.P., J.L., H.F.C., and H.Y. characterized the hydrogel device's in vitro properties and analyzed the data from in vivo and in vitro tests. C.S., N.V.P., J.C., and S.T. tested the in vivo behavior of the hydrogel device. S.L. performed the theoretical analysis of swelling of the hydrogel device. X.L., S.L., X.Z., G.A.P., C.S., and G.T. wrote the manuscript with input from all other authors. X.Z. and G.T. supervised the study.

## Additional information

**Competing interests:** X.Z., X.L., S.L., G.A.P., and H.Y. are co-inventors on provisional patent application number 62/623,695 describing the design of hydrogels with high swelling ratio, high swelling speed, and long-term robustness. G.T. is a co-inventor on multiple patents describing gastric-resident drug delivery systems and has a financial interest in Lyndra Inc. and Vivtex Inc., which are biotechnology companies focusing on the development of orally delivered systems for drug delivery and sensing via the gastrointestinal tract. All other authors declare no competing interests.

