## [Peer Review File · Nature Communications]

Reviewers' comments:

Reviewer #1 (Remarks to the Author):

The authors have prepared a fast swelling hydrogel system that can stay in the stomach for long period of time. The manuscript seems to be prepared to make the work fancier than what it actually is.

Title: A Pufferfish-Inspired Ingestible Hydrogel Machine

What is it about pufferfish-inspired? The necessity of fast swelling or deswelling of hydrogels has been known for decades, and it is not clear what pufferfish-inspired really means? Whether someone is inspired by pufferfish or not, the hydrogel described in the manuscript is already known.

Also, it is just a hydrogel, and why does it need to be called hydrogel machine? In this approach, all hydrogel based systems are all machines. Does this make any sense? Please do not try to appear fancy. A lot of scientists already new the fast swelling hydrogels.

Claim: “--- maintain robustness under repeated mechanical loads in the stomach for up to one month.” This may be the case for pigs, and it needs to be clarified. The pig data are not quite applicable to humans.

The claim of one-month residence time in the stomach has little meaning, as it is not associated with the drug release. It is noticed that the authors did not provide the drug release data. If a drug cannot be released for a month, what is the one-month residence time in the stomach for?

Overall, the study done are simply copy of what others have done, and it does not seem to add anything new or useful for developing truly gastric retentive devices. This manuscript needs to be revised to make only the scientific points by eliminating all fancy words, such as pufferfish-inspired and hydrogel machine.

Reviewer #2 (Remarks to the Author):

The manuscript reported the preparation of a fast-swelling hydrogel with high swelling ratio and its application as a soft gastric-retentive device for physiological monitoring. It is a very impressive work. I recommend the acceptance of this manuscript after the following questions are addressed.

1. What is the difference between the membrane with one freeze-thaw cycle and four freeze-thaw cycles?
2. The superabsorbent particles should be sodium polyacrylate homopolymers, Waster Lock 770. Polyacrylate homopolymers are different from sodium polyacrylate homopolymers.
3. In Figure S10, why do the authors choose 1.2 MPa as the maximal stress? The long-term mechanical force from the stomach is ~1000 cycles per day of 5-10 kPa.

Reviewer #3 (Remarks to the Author):

The authors prepared an ingestible hydrogel machine with prolonged retention time in stomach. Important properties including biocompatibility, mechanical compliance, robustness, fast swelling speed, big swelling ratio and controllable deswelling that are critical for prolonged retention and easy removal from stomach have been demonstrated. The design of encapsulating superabsorbent particles in anti-fatigue porous PVA membrane is new and successfully solves the common problems in existing systems such as slow swelling speed, limited swelling ratio and poor

mechanical integrity. The new system solves all the above mentioned problems at the same time. Additionally, the authors have demonstrated versatile capabilities of the system in multi-functionalities including incorporation of temperature sensors for monitoring and prolonged drug delivery. This manuscript demonstrated an innovative design and did systematic characterization. Publication is suggested. For further improvement, the authors are suggested to considering the following items.

- In caption of Figure 1 c, the description "Porous hydrogels swell in water with a low swelling ratio" is not accurate, many porous hydrogels with big swelling ratio have been reported. If the authors are comparing their results with specific hydrogel systems, please clarify by citing the specific references.
- There is mismatch of description in line 175: "we applied 1,920 cycles of 40% compressive strain on the hydrogel machine in SGF (pH 3) " and caption of Figure 3 f, line 530: "the hydrogel machine is immersed in water". Is the hydrogel machine immersed in water or SGF?

Response to the comments and suggestions from Reviewer #1

Comment 1. The authors have prepared a fast swelling hydrogel system that can stay in the stomach for long period of time. The manuscript seems to be prepared to make the work fancier than what it actually is.

Response 1. Thank you for your time in reviewing our paper. In order to address your and other reviewers' comments and suggestions, we have added a set of experiments, analyses and explanations. In the following paragraphs, we will address your comments point by point. We will also testify the novelty and significance of the current work. Sentences newly inserted into the manuscript and the supplementary information are marked in red.

Comment 2. Title: A Pufferfish-Inspired Ingestible Hydrogel Machine. What is it about pufferfish-inspired? The necessity of fast swelling or deswelling of hydrogels has been known for decades, and it is not clear what pufferfish-inspired really means?

Response 2. Pufferfish can quickly imbibe water (instead of diffusion) to inflate its body into a large ball, and maintain the imbibed water in a robust membrane (i.e., the skin of pufferfish) over the long term. The design of our hydrogel capsule is inspired by pufferfish, because the hydrogel capsule consists of (i) fast-swelling hydrogel particles that enable the whole capsule to quickly imbibe water (instead of diffusion) and (ii) a soft yet anti-fatigue hydrogel membrane that maintains long-term robustness of the hydrogel capsule in stomach. You can see that the design of our hydrogel capsule structure is indeed inspired by the (i) rapid water imbibing and (ii) robust skin of pufferfish. Our hydrogel device is the first design of such all-hydrogel capsule structure in the literature of hydrogels.

In addition, the necessity of fast swelling or deswelling of hydrogels has been known for decades, and there exist hydrogels that are either fast/highly swellable, or tough. But there exists no hydrogel that possesses the properties of fast-swelling, highly swellable, and robust over long term simultaneously, all of which are critical properties for the design of ingestible and long-term gastric-retentive hydrogel devices. This contrast between long-term development of the field and the lack of those desired properties indeed demonstrates the novelty and importance of our work.

To further clarify the relation of the design of our hydrogel capsule to pufferfish, we have added on Page 5, “**Here, we introduce a pufferfish-inspired hydrogel device, consisting of superabsorbent hydrogel particles that enables the device to quickly imbibe water (instead of diffusion) encapsulated in a soft yet anti-fatigue hydrogel membrane that maintains long-term robustness of the device.**”

Comment 3. Whether someone is inspired by pufferfish or not, the hydrogel described in the manuscript is already known.

Response 3. With due respect, we do not agree with this comment. To our knowledge, there exists no previous design of hydrogel structure that consists of a robust anti-fatigue hydrogel membrane encapsulating fast-swelling and highly swellable hydrogel particles.

In addition, the resultant design of our hydrogel capsule gives world-record performance in terms of combined swelling speed, ratio and anti-fatigue behaviors (i.e., swelling 100 times in 10 min, and sustaining 26,000 cycles of 20 N force), which is inaccessible in all previous swellable hydrogels made of single materials. The extraordinary properties of our hydrogel capsule were highly evaluated by other reviewers. For example, Reviewer #3 appreciated our work on the important properties as “*biocompatibility, mechanical compliance, robustness, fast swelling speed, big swelling ratio and controllable deswelling that are critical for prolonged retention and easy removal from stomach have been demonstrated*”, the innovative design as “*the design of encapsulating superabsorbent particles in anti-fatigue porous PVA membrane is new and successfully solves the common problems in existing systems such as slow swelling speed, limited swelling ratio and poor mechanical integrity*”.

In order to address this comment and clearly demonstrate the advance of our work in terms of hydrogel design, we have added **Supplementary Table 2** to compare existing hydrogels for ingestible applications in terms of mechanical and swelling properties. The table is also added to the Supplementary Information.

Supplementary Table 2. Comparison of mechanical and swelling performances of existing hydrogels for gastric retention. The red crosses represent the Achilles' heels of each hydrogel.

Hydrogels	Mechanical properties		Swelling properties		In vivo test	
	Young's modulus	Strength of swollen gel in SGF	Swelling ratio in volume	Swelling time	Gastric retention time	Animal model
Superporous hydrogels ^{12,19}	< 10 kPa	29 kPa ×	3.8-90	6 min	4-32 h	Dog
Stretchable superabsorbent hydrogels ⁴⁴	1 kPa	Not available	1000	> 1 day ×	Not available	Not available
Triggerable tough hydrogels ⁷	75 kPa	Not available	2.7 ×	6 days ×	7-9 days	Pig
pH responsive hydrogels ⁴³	7.24 MPa	~0 ×	10	1 day ×	2 day	Rabbit
Pufferfish-inspired hydrogel devices	3-10 kPa	70 kPa	20-100	10 min	9-29 days	Pig

It can be seen from the table that those previously reported hydrogels (refs. 7, 12, 19, 43, 44) exhibit very poor mechanical performances in simulated gastric fluid, due to the highly extended polymer networks, porous structures, and/or material degradation in acidic condition. Moreover, in the cases of hydrogels without porous structures (refs. 7, 43, 44), the swelling rates are severely limited by diffusion, thus requiring several days to swell.

Only the hydrogel device reported in our work can achieve the superior swelling and mechanical performances at the same time. In our work, we successfully solve the above-mentioned problems by encapsulating superabsorbent hydrogel particles within a robust hydrogel membrane to give the desirable swelling ratio and rate together with superior mechanical performances. Notably, Reviewer #3 commented on our work, “*The design of encapsulating superabsorbent particles in anti-fatigue porous PVA membrane is new and successfully solves the common problems in existing systems such as slow swelling speed, limited swelling ratio and poor mechanical integrity. The new system solves all the above mentioned problems at the same time.*”

We hope the novelty and importance of our pufferfish-inspired design of the hydrogel capsule is clear to the reviewer now.

Comment 4. Also, it is just a hydrogel, and why does it need to be called hydrogel machine? In this approach, all hydrogel based systems are all machines. Does this make any sense? Please do not try to appear fancy. A lot of scientists already new the fast swelling hydrogels.

Response 4. In the current paper, we explore the applications of our hydrogel capsule as an ingestible and gastric-retentive device capable of continuously measuring the gastric temperature of a pig model up to 29 days. To our knowledge, this is the first report on an ingestible and gastric-retentive device capable of one-month continuous measurement of gastric temperature. Because of its application as a device, we called the hydrogel capsule “hydrogel machine”.

From the reviewer’s **comment 4** and **comment 6**, it seems the reviewer neglected or misunderstood the application of the hydrogel capsule as a device. In the revised version of the paper, we have further emphasized the application of the capsule as a device capable of measuring the gastric temperature of a pig model up to 29 days and other potential applications (demonstrated in vitro). Moreover, we have changed the name in the title from “hydrogel machine” to “**hydrogel device**” to exactly reflect the device application of our hydrogel capsule.

In addition, while “a lot of scientists already new (knew) the fast swelling hydrogels”, none of previous fast-swelling hydrogels can maintain long-term robustness under mechanical loads (e.g., superporous hydrogels in refs. 12, 19). Similarly, none of existing fast-swelling hydrogel devices can reside in the stomach of a pig model for up to 29 days while continuously measuring the gastric temperature.

The unprecedented properties and functions of our new hydrogel validate the novelty and importance of the current work, which also has been highly appreciated by other reviewers. For

example, Reviewer # 2 praised “*this is a very impressive work*”, and Reviewer #3 highlighted our work on the important properties as “*biocompatibility, mechanical compliance, robustness, fast swelling speed, big swelling ratio and controllable deswelling that are critical for prolonged retention and easy removal from stomach have been demonstrated*”, the novel design as “*the design of encapsulating superabsorbent particles in anti-fatigue porous PVA membrane is new and successfully solves the common problems in existing systems such as slow swelling speed, limited swelling ratio and poor mechanical integrity*”, and the versatile capabilities as “*authors have demonstrated versatile capabilities of the system in multi-functionalities including incorporation of temperature sensors for monitoring and prolonged drug delivery*”.

In page 12-13, we added “**To our best knowledge, this is the first hydrogel-based device that achieves unprecedented properties of high swelling ratio, high swelling speed, and long-term robustness simultaneously. In addition, this is the first report on an ingestible and gastric-retentive device capable of one-month continuous measurement of gastric temperature in a pig model.**”

Comment 5. Claim: “--- maintain robustness under repeated mechanical loads in the stomach for up to one month.” This may be the case for pigs, and it needs to be clarified. The pig data are not quite applicable to humans.

Response 5. Thank you for the comment. Yorkshire pigs have gastric and intestinal anatomy and dimensions similar to humans, and have been widely used in the evaluation of structural integrity of other gastrointestinal devices (refs. 7, 10, 45). However, the interspecies differences with respect to gastric biomechanics still need to be taken into consideration, that is, the rate of gastric emptying and the gastric pressure in pigs tended to be smaller than those in humans (**Table R2**). Therefore, we recognized that the pig is a good model to test integrity of materials/devices in the gastric cavity, but not as good for retention. We have clearly identified the differences between pig model and human in the manuscript, page 10, “**one potential limitation of our *in vivo* tests is that pigs have slower gastric emptying than humans^{16,33}. Also, the gastric compression force in pigs are slightly lower in pigs than that in humans^{17,34}. For the successful translation to humans, further testing in other large animal species such as beagle dogs will likely be required³⁵.**”

Table R2. Comparison among gastrointestinal characteristics of pig, human, and dog (Refs. 34,45-51).

	Pig, 50 kg	Human, 70 kg	Dog, 10 kg
Capacity of stomach (ml)	2000-4000	1500	480
Basal acid output (ml/min)	1.05	1	0.3-1.5
Peak acid output (mEq/h)	5	18-23	39

Gastric pH in the fasting state	1.6-1.8	1.7	1.5
Periodicity of phase 3 (housekeeper wave)	> 1 day	1.76 h	1.90 h
Length of phase 3 activity	/	0.31 h	0.32 h
Gastric destructive pressure (kPa)	4	5-10	16
Total gastrointestinal transit time	1-10 days	20-30 h	6-8 h
Small intestinal transit time	~ 1 day	3 h	2 h
Duodenal diameter (mm)	30-50	30-40	20-25
Cut-off size for prolonged gastric retention (mm)	10-20	11-13	2-7
Size which does not empty from the stomach, e.g. foreign body (mm)	Larger than 45	Longer than 50 or larger than 30	Larger than 14

Comment 6. The claim of one-month residence time in the stomach has little meaning, as it is not associated with the drug release. It is noticed that the authors did not provide the drug release data. If a drug cannot be released for a month, what is the one-month residence time in the stomach for?

Response 6. As discussed in the response to comment 4, we focus on the application of our hydrogel capsules as a device capable of measuring gastric temperature over a long period of time. Although drug release is one of the most common applications that ingestible devices can demonstrated with, we tried to expand the hydrogel devices with other biomedical functions including body signal monitoring in the digestive system. The long-term GI tract body signals (~ one month) are previously inaccessible, though short-term (~ 24 h) data can obtained by letting pill-like capsules transit through the GI tract (ref. 3, 16). In addition, we did demonstrate the potential application of the hydrogel capsule for prolonged drug release with in vitro experiment (Supplementary Figure 1).

Comment 7. Overall, the study done are simply copy of what others have done, and it does not seem to add anything new or useful for developing truly gastric retentive devices. This manuscript needs to be revised to make only the scientific points by eliminating all fancy words, such as pufferfish-inspired and hydrogel machine.

Response 7. We do not agree with this comment. In the above point-to-point responses, we have explained the novelty and significance of our hydrogel device in terms of unprecedented properties and long-term body signal monitoring.

In addition, we have replaced the “hydrogel machine” with “hydrogel device” to exactly reflect its device application, and revised the language in the whole manuscript to make it clearer. We further added more discussions on the information inferred from the prolonged and continuous gastric temperature, which is potentially important for monitoring of the behavior pattern, analysis of the circadian rhythm, and diagnosis of abnormality.

We hope the extensive revision and explanation can satisfactorily address your comments and concerns on the paper. Thank you once again for your time in reviewing the paper.

Response to the comments and suggestions from Reviewer #2

General comment. The manuscript reported the preparation of a fast-swelling hydrogel with high swelling ratio and its application as a soft gastric-retentive device for physiological monitoring. It is a very impressive work. I recommend the acceptance of this manuscript after the following questions are addressed.

Response. Thank you very much for pointing out the novelty and significance of our work. We also greatly appreciate your insightful suggestions and comments, which help us to further strengthen the paper.

Comment 1. What is the difference between the membrane with one freeze-thaw cycle and four freeze-thaw cycles?

Response 1. Thank you for the question. The crystallinity of PVA hydrogel is increased with freeze-thaw cycle numbers. Thus, the mechanical properties (including Young's modulus, toughness, fatigue threshold and others) are varied over freeze-thaw cycles (shown in Table R3) (ref. 32). In particular, the PVA hydrogel membrane with four freeze-thaw cycles has a higher fatigue resistance than the membrane with one freeze-thaw cycle, while it maintains the relatively low modulus, leading to a mechanically soft and robust hydrogel device with a high swelling ratio (Figure 2d).

Table R3. Comparison of PVA hydrogels with one freeze-thaw cycle and four freeze-thaw cycles (data come from the work under revision, Lin et al., 2018).

PVA hydrogels	Crystallinity in dry state	Young's modulus (kPa)	True strength (MPa)	Fatigue threshold (J/m ²)
One freeze-thaw cycle	2%	3	0.12	29
Four freeze-thaw cycles	37%	47	8.2	310

Comment 2. The superabsorbent particles should be sodium polyacrylate homopolymers, Waster Lock 770. Polyacrylate homopolymers are different from sodium polyacrylate homopolymers.

Response 2. Thank you for the rigorous suggestion. We corrected the chemical name of superabsorbent particles into **sodium polyacrylate homopolymers**.

Comment 3. In Figure S10, why do the authors choose 1.2 MPa as the maximal stress? The long-term mechanical force from the stomach is ~ 1000 cycles per day of 5-10 kPa.

Response 3. Thank you for raising this question. It is true that the gastric mechanical pressure on the ingestible device (e.g., food or hydrogel device) is ~ 5 -10 kPa (ref. 17). However, the stress experienced by the encapsulating membrane is much higher than the compressive stress experienced by the whole structure of hydrogel device, due to the following reasons:

- (1) At the undeformed state of hydrogel device (for example, Figure 2a, fully swollen hydrogel device), its expanded membrane has already been stretched biaxially owing to the inflated core materials;
- (2) By further compressing the swollen hydrogel device, the hydrogel membrane experiences even larger deformation and larger stresses, and the deformation in the hydrogel membrane are not uniform and highly localized at the side parts that are not directly contact with the clamps (Figure 3d);
- (3) There is significant difference of fatigue behavior between polyacrylamide-agar hydrogel and PVA hydrogel. We showed in supplementary information that the polyacrylamide-agar hydrogel cannot sustain 1.2 MPa after 110 cycles of tensile loading, while PVA can (Supplementary Figure 10, b and c). In the revision, we added that the sustainable stress on PVA membrane can be achieved as high as 4.3 MPa after 9,000 cycles (Supplementary Figure 10d). Therefore, the polyacrylamide-agar hydrogel has low sustainable stress and short fatigue life, while PVA hydrogel exhibit dramatically enhanced fatigue-resistance;
- (4) The real gastric environment can be more complicated and demanding for long-term gastric retention. Actually before we testified the hydrogel device with PVA antifatigue hydrogel membrane, we used the tough hydrogel membrane (polyacrylamide-agar hydrogel), which easily ruptures in the porcine stomach within two day (Figure R1, a failed in vivo experiment), indicating that the common tough hydrogel cannot sustain the real gastric contraction. After that, we turned to the fatigue-resistant hydrogel membrane, that can be applicable in the real stomach.

Figure R1. Short stay in the pig stomach of hydrogel device with commonly used tough hydrogel membrane (polyacrylamide-agar hydrogel). X-ray images of the hydrogel device residing in the pig stomach for only two days. N = 1 for the test. Scale bar is 5 cm.

Supplementary Figure 10. Comparison of long-term strength under cyclic tensile test between two hydrogel membranes. (a) The undeformed state (left) and the deformed state (right) of the porous antifatigue PVA hydrogel membrane. The sample remains stable after 1, 1000, and 10,000 cycles of tensile loading. (b) The PVA hydrogel membrane can sustain stress of 1.2 MPa for 10,000 cycles of tensile loading. (c) The polyacrylamide-agar hydrogel can reach the maximum stress of 1.2 MPa but ruptures within 110 cycles of tensile loading. (d) The PVA hydrogel membrane can sustain stress of 4.3 MPa for 9,000 cycles of tensile loading. The strain rates are set as 5 mm/s. $N = 3$ for each test. Scale bars are 5 mm for a.

Response to the comments and suggestions from Reviewer #3

General comment. The authors prepared an ingestible hydrogel machine with prolonged retention time in stomach. Important properties including biocompatibility, mechanical compliance, robustness, fast swelling speed, big swelling ratio and controllable deswelling that are critical for prolonged retention and easy removal from stomach have been demonstrated. The design of encapsulating superabsorbent particles in anti-fatigue porous PVA membrane is new and successfully solves the common problems in existing systems such as slow swelling speed, limited swelling ratio and poor mechanical integrity. The new system solves all the above mentioned problems at the same time. Additionally, the authors have demonstrated versatile capabilities of the system in multi-functionalities including incorporation of temperature sensors for monitoring and prolonged drug delivery. This manuscript demonstrated an innovative design and did systematic characterization. Publication is suggested. For further improvement, the authors are suggested to considering the following items.

Response. Thank you very much for summarize the novelty and significance of our work. We also greatly appreciate your insightful suggestions and comments, which help us to further strengthen the paper.

Comment 1. In caption of Figure 1c, the description “Porous hydrogels swell in water with a low swelling ratio” is not accurate, many porous hydrogels with big swelling ratio have been reported. If the authors are comparing their results with specific hydrogel systems, please clarify by citing the specific references.

Response 1. Thanks for pointing this out. Indeed, there are several types of porous hydrogel with high swelling ratio. For example, the superporous polyacrylate hydrogels, where the electrostatic repulsive forces between negatively charged polymer chains highly increased the equilibrium swelling ratio (ref. 19). We recognize the superporous hydrogels with both fast and large swelling in the comparison chart (Figure 2c).

However, in the comparison of figure 1, we only focus on the hydrogels made of neutral polymer networks with negligible charge effects (ref. 18 as an example for porous gel, ref. 7 as an example for bulk hydrogel, and the PVA hydrogel membrane in the hydrogel device). We exclude the highly charged hydrogel and only focus on the neutral hydrogel, since highly charged hydrogel alone is extremely brittle and weak in swollen state. In order to specify this comparison, we add the references^{7,18} in the Figure 1b and c, respectively.

Comment 2. There is mismatch of description in line 175: “we applied 1,920 cycles of 40% compressive strain on the hydrogel machine in SGF (pH 3)” and caption of Figure 3 f, line 530: “the hydrogel machine is immersed in water”. Is the hydrogel machine immersed in water or SGF?

Response 2. Thank you for the correction. The hydrogel device should be “immersed in SGF (pH 3)” in Figure 3f, and the caption in line 530 has been corrected. We also provide the data of hydrogel device under cyclic compression in water in the supplementary figure 9.

References (1-42 are also listed in the manuscript)

1. Someya, T., Bao, Z. & Malliaras, G.G. The rise of plastic bioelectronics. *Nature* 540, 379 (2016).
2. Bettinger, C.J. Materials advances for next-generation ingestible electronic medical devices. *Trends in biotechnology* 33, 575-585 (2015).
3. Kalantar-Zadeh, K., et al. A human pilot trial of ingestible electronic capsules capable of sensing different gases in the gut. *Nature Electronics* 1, 79 (2018).
4. Mimee, M., et al. An ingestible bacterial-electronic system to monitor gastrointestinal health. *Science* 360, 915-918 (2018).
5. Nieben, O.G. & Harboe, H. Intra-gastric balloon as an artificial bezoar for treatment of obesity. *The Lancet* 319, 198-199 (1982).
6. Lee, Y., et al. Therapeutic luminal coating of the intestine. *Nature Materials* (2018).
7. Liu, J., et al. Triggerable tough hydrogels for gastric resident dosage forms. *Nature communications* 8, 124 (2017).
8. Bellinger, A.M., et al. Oral, ultra-long-lasting drug delivery: application toward malaria elimination goals. *Science translational medicine* 8, 365ra157-365ra157 (2016).
9. Singh, B.N. & Kim, K.H. Floating drug delivery systems: an approach to oral controlled drug delivery via gastric retention. *Journal of Controlled release* 63, 235-259 (2000).
10. Zhang, S., et al. A pH-responsive supramolecular polymer gel as an enteric elastomer for use in gastric devices. *Nature materials* 14, 1065 (2015).
11. Kagan, L., et al. Gastroretentive accordion pill: enhancement of riboflavin bioavailability in humans. *Journal of controlled release* 113, 208-215 (2006).
12. Chen, J., Blevins, W.E., Park, H. & Park, K. Gastric retention properties of superporous hydrogel composites. *Journal of Controlled Release* 64, 39-51 (2000).
13. Drury, J.L. & Mooney, D.J. Hydrogels for tissue engineering: scaffold design variables and applications. *Biomaterials* 24, 4337-4351 (2003).
14. Liu, X., et al. Stretchable living materials and devices with hydrogel-elastomer hybrids hosting programmed cells. *Proceedings of the National Academy of Sciences* 114, 2200-2205 (2017).
15. Salessiotis, N. Measurement of the diameter of the pylorus in man: Part I. Experimental project for clinical application. *The American Journal of Surgery* 124, 331-333 (1972).
16. Koziolk, M., et al. Investigation of pH and temperature profiles in the GI tract of fasted human subjects using the Intellicap® system. *Journal of pharmaceutical sciences* 104, 2855-2863 (2015).
17. Houghton, L., et al. Motor activity of the gastric antrum, pylorus, and duodenum under fasted conditions and after a liquid meal. *Gastroenterology* 94, 1276-1284 (1988).
18. Sun, B., Wang, Z., He, Q., Fan, W. & Cai, S. Porous double network gels with high toughness, high stretchability and fast solvent-absorption. *Soft matter* 13, 6852-6857 (2017).
19. Chen, J., Park, H. & Park, K. Synthesis of superporous hydrogels: hydrogels with fast swelling and superabsorbent properties. *Journal of Biomedical Materials Research: An Official Journal of The Society for Biomaterials, The Japanese Society for Biomaterials, and the Australian Society for Biomaterials* 44, 53-62 (1999).
20. Wainwright, P.C., Turingan, R.G. & Brainerd, E.L. Functional morphology of pufferfish inflation: mechanism of the buccal pump. *Copeia*, 614-625 (1995).
21. Kirti & Khora, S.S. Mechanical properties of pufferfish (*Lagocephalus gloveri*) skin and its collagen arrangement. *Marine and freshwater behaviour and physiology* 49, 327-336 (2016).

22. Berens, A. & Hopfenberg, H. Diffusion and relaxation in glassy polymer powders: 2. Separation of diffusion and relaxation parameters. *Polymer* 19, 489-496 (1978).
23. Ross, A.C., Taylor, C.L., Yaktine, A.L. & Del Valle, H.B. Tolerable Upper Intake Levels: Calcium and Vitamin D. (2011).
24. Sultan, M. & Norton, R.A. Esophageal diameter and the treatment of achalasia. *The American journal of digestive diseases* 14, 611-618 (1969).
25. Trande, P., et al. Efficacy, Tolerance and Safety of New Intra-gastric Air-Filled Balloon (Heliosphere BAG) for Obesity: the Experience of 17 Cases. *Obesity Surgery* 20, 1227-1230 (2010).
26. Cheifetz, A.S., et al. The risk of retention of the capsule endoscope in patients with known or suspected Crohn's disease. *The American journal of gastroenterology* 101, 2218 (2006).
27. Mortensen, L. & Charles, P. Bioavailability of calcium supplements and the effect of Vitamin D: comparisons between milk, calcium carbonate, and calcium carbonate plus vitamin D. *The American journal of clinical nutrition* 63, 354-357 (1996).
28. Thomas, R., Yeoh, T., Wan-Nadiah, W. & Bhat, R. Quality evaluation of flat rice noodles (Kway Teow) prepared from Bario and Basmati rice. *Sains Malaysiana* 43, 339-347 (2014).
29. Koza, H., et al. Development of a human gastric digestion simulator equipped with peristalsis function for the direct observation and analysis of the food digestion process. *Food Science and Technology Research* 20, 225-233 (2014).
30. McKee, C.T., Last, J.A., Russell, P. & Murphy, C.J. Indentation versus tensile measurements of Young's modulus for soft biological tissues. *Tissue Engineering Part B: Reviews* 17, 155-164 (2011).
31. Bai, R., et al. Fatigue fracture of tough hydrogels. *Extreme Mechanics Letters* 15, 91-96 (2017).
32. Hassan, C.M. & Peppas, N.A. Structure and applications of poly (vinyl alcohol) hydrogels produced by conventional crosslinking or by freezing/thawing methods. in *Biopolymers: PVA Hydrogels, Anionic Polymerisation Nanocomposites* 37-65 (Springer, 2000).
33. Snoeck, V., et al. Gastrointestinal transit time of nondisintegrating radio-opaque pellets in suckling and recently weaned piglets. *Journal of controlled release* 94, 143-153 (2004).
34. Houpt, T.R. Gastric pressures in pigs during eating and drinking. *Physiology & behavior* 56, 311-317 (1994).
35. Cargill, R., et al. Controlled gastric emptying. II. In vitro erosion and gastric residence times of an erodible device in beagle dogs. *Pharmaceutical research* 6, 506-509 (1989).
36. Sauve, C.C., Van de Walle, J., Hammill, M.O., Arnould, J.P. & Beauplet, G. Stomach temperature records reveal nursing behaviour and transition to solid food consumption in an unweaned mammal, the harbour seal pup (*Phoca vitulina*). *PLoS one* 9, e90329 (2014).
37. Thouzeau, C., Peters, G., Le Bohec, C. & Le Maho, Y. Adjustments of gastric pH, motility and temperature during long-term preservation of stomach contents in free-ranging incubating king penguins. *Journal of experimental biology* 207, 2715-2724 (2004).
38. Strong, J., Cameron, D. & Riddell, M. The electrolyte concentration of human gastric secretion. *Experimental physiology* 45, 1-11 (1960).
39. Pharmacopeia, U. Simulated gastric fluid, TS. *The National Formulary* 9(2000).
40. Gent, A., Lindley, P. & Thomas, A. Cut growth and fatigue of rubbers. I. The relationship between cut growth and fatigue. *Journal of Applied Polymer Science* 8, 455-466 (1964).
41. Johnson, K.L. & Johnson, K.L. *Contact mechanics*, (Cambridge university press, 1987).

42. Zhang, L., Bailey, J.B., Subramanian, R.H. & Tezcan, F.A. Hyperexpandable, self-healing macromolecular crystals with integrated polymer networks. *Nature* 557, 86 (2018).
43. Wu, T., et al. A pH-Responsive Biodegradable High-Strength Hydrogel as Potential Gastric Resident Filler. *Macromolecular Materials and Engineering* 1800290 (2018).
44. Cipriano, B. H. et al. Superabsorbent hydrogels that are robust and highly stretchable. *Macromolecules* 47, 4445-4452 (2014).
45. Swindle, M., Makin, A., Herron, A., Clubb Jr, F. & Frazier, K. Swine as models in biomedical research and toxicology testing. *Veterinary pathology* 49, 344-356 (2012).
46. Aoyagi, N. et al. Gastric emptying of tablets and granules in humans, dogs, pigs, and stomach-emptying-controlled rabbits. *Journal of pharmaceutical sciences* 81, 1170-1174 (1992).
47. Klausner, E. A., Lavy, E., Friedman, M. & Hoffman, A. Expandable gastroretentive dosage forms. *Journal of controlled release* 90, 143-162 (2003).
48. Kararli, T. T. Comparison of the gastrointestinal anatomy, physiology, and biochemistry of humans and commonly used laboratory animals. *Biopharmaceutics & drug disposition* 16, 351-380 (1995).
49. Hossain, M., Abramowitz, W., Watrous, B. J., Szpunar, G. J. & Ayres, J. W. Gastrointestinal transit of nondisintegrating, nonerodible oral dosage forms in pigs. *Pharmaceutical research* 7, 1163-1166 (1990).
50. Treacy, P., Jamieson, G. & Dent, J. Pyloric motility and liquid gastric emptying during barostatic control of gastric pressure in pigs. *The Journal of physiology* 474, 361-366 (1994).
51. Kim, Y., Yuk, H., Zhao, R., Chester, S. A. & Zhao, X. Printing ferromagnetic domains for untethered fast-transforming soft materials. *Nature* 558, 274 (2018).

REVIEWERS' COMMENTS:

Reviewer #2 (Remarks to the Author):

The authors have addressed all our questions.

Editorial Note: Reviewer #2 was asked to evaluate the author responses to Reviewer #1's comments. Reviewer #2 informed the editor that all of Reviewer #1's comments had been addressed adequately.

REVIEWERS' COMMENTS:

Reviewer #2 (Remarks to the Author):

The authors have addressed all our questions.

Response. Thank you for taking time to review our manuscript.